# Mapping the Scientific Research on Healthcare Workers’ Occupational Health: A Bibliometric and Social Network Analysis

**DOI:** 10.3390/ijerph17082625

**Published:** 2020-04-11

**Authors:** Bingke Zhu, Hao Fan, Bingbing Xie, Ran Su, Chaofeng Zhou, Jianping He

**Affiliations:** 1School of Public Health and Management, School of Humanities and Social Science, Hubei University of Medicine, Shiyan 442000, China; echo@hbmu.edu.cn (B.Z.); suranryan@163.com (R.S.); zcf1990@gmail.com (C.Z.); 2School of Information Management, Wuhan University, Wuhan 430000, China; jianpinghe@whu.edu.cn

**Keywords:** healthcare workers, occupational health, bibliometric, SNA, occupational exposure, physical injuries, infectious diseases

## Abstract

In the last few years, the occupational health (OH) of healthcare workers (HCWs) has been shown increasing concern by both health departments and researchers. This study aims to provide academics with quantitative and qualitative analysis of healthcare workers’ occupational health (HCWs+OH) field in a joint way. Based on 402 papers published from 1992 to 2019, we adopted the approaches of bibliometric and social network analysis (SNA) to map and quantify publication years, research area distribution, international collaboration, keyword co-occurrence frequency, hierarchical clustering, highly cited articles and cluster timeline visualization. In view of the results, several hotspot clusters were identified, namely: physical injuries, workplace, mental health; occupational hazards and diseases, infectious factors; community health workers and occupational exposure. As for citations, we employed document co-citation analysis to detect trends and identify seven clusters, namely tuberculosis (TB), strength training, influenza, healthcare worker (HCW), occupational exposure, epidemiology and psychological. With the visualization of cluster timeline, we detected that the earliest research cluster was occupational exposure, then followed by epidemiology and psychological; however, TB, strength training and influenza appeared to gain more attention in recent years. These findings are presumed to offer researchers, public health practitioners a comprehensive understanding of HCWs+OH research.

## 1. Introduction

In April 2019, the International Labour Organization (ILO) declared a mortality of nearly three million workers worldwide due to occupation-related diseases or accidents on an annual basis, and with another 374 million people injured in occupation-related accidents, which is “unacceptable loss of life” [1], as officials put it. Worldwide, healthcare workers (HCWs) make up 12% of the workforce [2] and face high occupational risk. In December 2019, Dr. Yang Wen of Beijing Civil Aviation General Hospital passed away as a result of a severe violent injury inflicted by a patient, this aroused public concern for the occupational health (OH) of HCWs. A healthcare worker (HCW) is defined as any individual whose work is in a medical environment; it includes but is not limited to doctors, nurses, nursing assistants, operation theater technicians, community healthcare workers, pharmacists, etc. [3]. However, HCWs who fight for others’ health may encounter risks from strenuous manual labor, infectious diseases, chemical hazards, and irrational patients, which has attracted the attention of society, scholars, public health practitioners, and healthcare managers [4,5].

Studies on healthcare workers’ occupational health (HCWs+OH) are conducted from diverse perspectives. Infectious factors and musculoskeletal injuries caused while shifting patients have attracted the attention of employee OH programs thus far. Yip et al. [6] and Petit et al. [7] summarized that the prevalence of musculoskeletal injury (MSI) was high among HCWs, and those who often manually carry patients had a higher risk of injury than those who did not. It is inevitable for HCWs to expose themselves to patients’ blood as well other body fluids (OEB) which may be caused by direct contact with patients who suffer from needlestick injuries [8] and sharp object damage [9]. Mucocutaneous engagement with the body fluids of the patients [10] could lead to the potential of catching infectious diseases, hepatitis B virus (HBV) and hepatitis C virus (HCV) for instance [11]; human immunodeficiency virus (HIV) [12], severe acute respiratory syndrome (SARS) [13], Ebola virus [14] and tuberculosis (TB) are also cases in point [15,16]. Patz et al. [17] put forward that surgeons, obstetricians and gynecologists were the group with the highest exposure rate. Chiodi et al. [18] and Oltulu et al. [19] discussed that industry hazards, biopharma products and other chemical hazards had been recognized as risk factors for HCWs. Additionally, natural rubber latex (NRL) products, especially examination gloves, were frequently employed in the medical environment [20]. Verna et al. [21] concluded that latex allergy had become an occupational hazard for HCWs; furthermore, allergies and exposure levels could be seen as predisposing factors for latex sensitization.

In the last few years, it is clear that workplace violence has been considered as a severe problem for HCWs. It has been defined by the health sector as a violent attack on workers in the general field [22,23,24]. Workplace violence can potentially cause mental problems like depression, inferiority and demotivation [25,26,27]. Ito et al. [28] conducted one analysis of the Brief Job Stress Questionnaire and found that the "total health risk" of HCWs was 10% higher than the national average. Due to organizational factors and the deficiency in social support at work, HCWs may experience occupational stress, which may cause burnout or psychosomatic diseases. Demou et al. [29] concluded that mental health conditions, depression particularly, were a prominent cause of the absence of many HCWs. Moreover, reflecting the prevalence of chronic diseases and increasing health care costs, more people were receiving treatment domestically, which would largely increase the requirement for home healthcare workers (HHCWs) [30]. Increasing work intensity, adverse working conditions, motor vehicle traffic and precarious work arrangements together contributed to high injuries for HHCWs [31,32,33].

However, publication time and number of related studies indicate that existing research has failed to carry out a systematic and comprehensive analysis of the rapid development of HCWs+OH research in the past. Meanwhile, it is no easy job to integrate existing research orientations and establish a systematic and comprehensive knowledge structure for HCWs+OH. Therefore, dedication to the overall progress of HCWs+OH objectively and quantitatively is quite significant both in theory and in practice. This study intends to analyze HCWs+OH research during 1992–2019 via bibliometric and social network analysis (SNA) method, comprehensively, with aims as follows: (1) identify the evolution and characteristic of research outputs, research area distribution, international collaboration, author productivity and collaboration; (2) highlight the hotspots of the HCWs+OH field based on keyword co-occurrence frequencies and hierarchical cluster analysis; (3) track the trend via per-document co-citation, highly cited articles, and cluster timeline visualization. This study is expected to provide researchers, public health practitioners, and healthcare managers with a systematic and comprehensive understanding of HCWs+OH research and a reference for further research as well.

## 2. Materials and Methods

### 2.1. Data Sources

Data illustrated here were retrieved from the Web of Science database Core Collection (WOSCC) provided by Thomson Reuters including the Science Citation Index Expanded (SCI-EXPANDED), Arts & Humanities Citation Index (A&HCI), Social Sciences Citation Index (SSCI), Conference Proceedings Citation Index-Science (CPCI-S), Conference Proceedings Citation Index-Social Science & Humanities (CPCI-SSH), and Emerging Sources Citation Index (ESCI). WOS is probably the most commonly used bibliometric analysis database for its accessibility to powerful web data on bibliography and citations which covers more than 12,000 highly acclaimed influential journals around the world [34]. To ensure the quality of data, this study limited the types of documents to articles, excluding conference proceedings, book chapters, letter or editorial material. For assurance of the comprehensiveness of the search results, a synonym search of HCWs+OH was realized by browsing the bibliographic information. The advanced search strategy of retrieving the topic on HCWs+OH was as follows: (TS = “medical worker *” OR TS = “health worker*” OR TS = “healthcare worker*” OR TS = “doctors and nurses” OR TS = “medical stuff” OR TS = “hospital stuff”) AND (TS = “occupation* health” OR TS = “professional health” OR TS = “occupational injury” OR TS = “occupational damage” OR TS = “self health”). The update of the WOSCC database was considered in the retrieval, with the time span being limited to 1992–2019. Eventually, a total of 402 academic journal articles were retrieved from the WOSCC database at the retrieval time of May 13, 2019, and processed by the Notepad++ software (developed by Don Ho, a free source code editor) initially.

### 2.2. Analysis Methods

Bibliometrics is a widely used means to evaluate and quantify the characteristics and development of specific disciplines [35], which adopts mathematical and statistical methods to quantify the characteristics, structures, relationships, and patterns of literature [36]. The main aspects of conducting a bibliometric analysis include Performance Analysis and Science Mapping [37]. Performance Analysis is based on the analysis of bibliometrics and evaluates the types, publication years, journals, research areas, countries and other characteristics of publications. Science Mapping is designed to indicate the structure of the research field. By using the Science Mapping tool CiteSpace (developed by Chaomei Chen, Drexel University, PA, USA), which can be used to visualize the knowledge structure in specific areas [38], this study analyzed the international collaboration and document co-citation to present research collaboration network and research trends of HCWs+OH. In order to more accurately predict the trend of the publication years, the time series method was also adopted. The time series method is to analyze time series data with calculated data in unified time space as foundation, to find out the significant statistics and properties of data and predict future events [39].

SNA works as a rather standardized analysis method in the field of social relations and social structure by analyzing the network relationships displayed with nodes and ties [40,41]. Keyword co-occurrence analysis functions as an effective way to track research topics, as keywords provide a high-level summary of the article which enables researchers to acquire deeper comprehension of the current situation in specific fields [42]. In this study, BibExcel [43] was applied to analyze each paper. The bibliography was used to calculate authors’ productivity and keyword co-occurrence matrix. Pajek [44] was used to generate an author collaboration network and calculate the centralities, namely degree centrality (DC), betweenness centrality (BC) and closeness centrality (CC). The measurement of a node’s centrality represents the importance of the node in the network. The BC measures the node’s ability to connect with other nodes in the network, while CC measures how easy it is for a node to reach other nodes, and DC measures how many directly linked nodes a node has [45]. UCINET (Analytic Technologies, Kentucky, USA) was employed to manage keywords co-occurrence matrices, and NetDraw (Analytic Technologies, Kentucky, USA) was another application for the generating of keyword co-occurrence network [43]. In addition, the statistical software SPSS (IBM, New York, USA) was used to generate keyword co-occurrence hieratical clusters, which can make further efforts to indicate the present research hotspots of HCWs+OH.

## 3. Results

### 3.1. Chronological Publication Trend

Figure 1 explains the yearly distribution of publications on HCWs+OH and its percentage among the total articles in the field of OH in WOSCC per year. During the past 28 years, the number of publications on HCWs+OH showed an up-going trend. According to the number of publications, research on HCWs+OH can be roughly divided into three consecutive stages. The starting period from 1992 to 2006 was quite stable, with the annual number of publications being basically the same (on average, five articles per year). There were only two articles (1.47%) published in 1992. In the 2007 year, demarcated as the debut of the second interval, the number of publications increased to 16 articles, and for the following years the publication maintained an average of 21 articles till 2014. The last period showed the greatest increase, starting from 2015 with 36 articles (3.55%) published, and the number of publications reached a peak of 43 (3.86%) in 2016, with an average of 41 articles per year from 2015 to 2018.

We employed the ARIMA (Autoregressive Integrated Moving Average Model) to predict the trend of publication on HCWs+OH and its percentage of the OH research field for the next ten years (i.e., 2019–2028). ARIMA is a comprehensive analysis technique that combines autoregressive and moving average analysis techniques, which provide a uncomplicated yet effective way for skilled time series prediction [46]. The publication percentage can be considered as an indicator of future topic prevalence in the OH research field. According to the ARIMA model, research on HCWs+OH may continue to increase in the next decade. It was predicted that the number of articles related to HCWs+OH may reach 87 in 2027, accounting for 4.10% of OH research per year. It is suggested that the field of HCWs+OH will continue to attract growing attention from multidisciplinary research communities.

### 3.2. Research Areas Distribution

All the articles are distributed in 58 identified research areas in the WOSCC database. As shown in Figure 2, the top 10 research fields include public environmental occupational health (192 articles; accounting for 47.46% of the total); infectious diseases (53; 13.18%); general internal medicine (46; 11.44%); nursing (37; 9.2%); health care sciences services(25; 6.22%); immunology(17; 4.23%); psychology(11;2.74%); microbiology (9; 2.24%); rehabilitation (9; 2.24%); and emergency medicine (8; 1.99%).

### 3.3. International Productivity and Collaboration Analysis

The network of international collaboration from 1992 to 2019 consists of 21 nodes and 15 links (Figure 3); the 10 countries with the most publications are listed in Table 1. In Figure 3, the size of the node stands for the number of articles issued by the country, and the thickness of the connection represents the strength of cooperation between countries [38]. As shown in Figure 3, America makes the most contributions, publishing 83 articles, followed by Canada (42). The third is Brazil (39) in South America. Four prominent nodes in Europe include England (30), France (12), Spain (11), and the Netherlands (11) ranked fourth, seventh, eighth and ninth. South Africa ranks fifth with 18 articles published. One node in Asia is China (11), with tenth rank.

According to Table 1, America accounts for 60.74% of the total output in the top ten countries, Europe for 24.44%. Moreover, in the light of the higher BC of the USA (0.41), Canada (0.33), and Brazil (0.31), they play vital roles in establishing links with other countries. South Africa and China account for 6.67% and 4.07% respectively. There were 11 articles published in China, but the BC is 0. To some extent, this indicates a lack of cooperation between China and other countries in HCWs+OH field.

### 3.4. Author Productivity and Collaboration

From BibExcel (developed by Olle Persson, infork, Umea University) statistics, 402 articles were retrieved including 1964 authors in total, which means 0.2 articles per author. Nearly 82.28% of the authors published one article, and 5.01% published two; it is evident that there are many more authors who have published at least an article in this field. The author co-occurrence network can be used to revel the cooperation between authors, which can help identify author groups [47]. Only a minor number of 38 published more than three papers among these 1964 authors. Pajek, which helps to identify the important authors, was used to calculate the DC, BC, CC and construct the author co-occurrence network. Table 2 shows the significant authors identified by multiple measures of centrality and ranked by number of publications.

As shown in Table 2, the most productive author was Yassi A with 17 articles, followed by Alamgir H with 9, Spiegel JM (9), Jakobsen MD (8), Andersen LL (8), Bryce EA (8), Brandt M (7), Sundstrup E (7), Yu SC (7), MacIntyre CR (5) and Seale H (5). Jay K (4), Aagaard P (4), Zungu M (4), Rahman B (4), Lockhart K (4), Zhang Y (3), Sheehan C (3), Donohue R (3) and Wang QY (3) published less than five papers. As shown in Figure 4, Yassi A, who had the most publications, was also the top author by centrality measures with the DC of 7, BC of 0.05, CC of 0.28. Alamgir H, who published nine papers, had a DC of 4, BC of 0.02, CC of 0.22. Yu SC, who published seven papers, had a DC of 4, BC of 0.02, CC of 0.22. Other authors who published more than three articles had BC of 0, indicating there being fewer direct relationships links with other authors.

Figure 4 shows the collaboration network of the top 20 most productive authors who had published more than three papers. In Figure 4, each vertex stands for one author, the size of the vertex is based on DC and the thickness of interconnecting lines signifies the symbol of the strength of collaboration [48]. According to Figure 4, Yassi A, Alamgir H, Spiegel JM, Yu SC, Ngan K, Drebit S, Zungu M, Bryce EA, Lockhart K, Nophale L were the most collective authors, and Yassi A took central position in the collaborative network. Jakobsen MD, Aagaard P, Andersen LL, Sundstrup E, Brandt M and Jay K had high collaboration. Wang QY, Chughtai AA, MacIntyre CR, Rahman B, Zhang Y and Seale H had higher collaboration, and Seale H took the central position. Shea T, De Cieri H, Cooper B, Sheehan C and Donohue R collaborated with each other, and Donohue R took the central position.

### 3.5. Research Hotspot of HCWs+OH

#### 3.5.1. Keywords Co-Occurrence Frequency Analysis

As a part of periodical papers, keywords can offer readers brief information about the research topic or method of the paper, and keywords co-occurrence analysis is often employed to detect the direction and hotspots in the research field [49]. BibExcel isolated the keyword information by applying filters in the metadata provided by WOSCC, and then generated a keyword co-occurrence matrix [50]. The keywords co-occurrence matrix was introduced into UCINET to generate a friendly format that can be used in NetDraw for drawing the keyword co-occurrence network map.

A total of 1730 individual keywords were adopted in 402 articles which were retrieved from WOSCC. However, 830 (47.96%) appeared just once and 1267 (73.24%) appeared less than seven times. Table 3 shows the other 26.76% of the total keywords that appeared at least seven times. Except for “OH (152)”, “HCW (85)” and “health personal (17)”, the most frequently appeared keywords were “TB (20)”, “infection control (18)”, “HIV (16)”, “vaccination (13)”, “influenza (11)” and “epidemiology (8)”, which indicates the impact of infectious diseases on HCWs. Besides, “occupational injury (14)”, “occupational exposure (13)”, “stress (12)”, “mental health (8)”, “back pain (7)”, “musculoskeletal disorders (7)”, and “working conditions (7)” also presented in the top 22 most frequently used keywords.

BibExcel was used to process the high frequency vocabulary to generate a 51 × 51 keyword co-occurrence matrix. Figure 5 shows the co-occurrence network of the 51 most frequently appeared keywords which showed up at least four times. Each vertex represents a keyword, and the size based on the thickness and the centrality of the lines indicates the strength of the connection [48]. The arrow indicates pointing to a relationship of positive correlation; the denser the arrow is, the stronger the centrality of the node is and the more important it is in the network. Evidently, “OH” and “HCW” are the most central keywords in all publications. Several other keywords are placed in the kernel of the network and the following are hotspots that drew much attention: “infection control”, “epidemiology”, “HIV”, “influenza”, “occupational exposure”, “occupational injury”, “primary healthcare”, and “mental health”, “workplace”. Additionally, “safety”, “stress”, “working conditions”, “surveillance”, “personal protective equipment”, “sickness absence”, “job satisfaction” “burnout”, “South Africa”, “TB”, “AIDS”, “vaccination”, “needlestick injuries”, “blood-borne pathogens”, “strength training”, “surgery”, “latex allergy” are also the research hotspots in HCWs+OH field.

#### 3.5.2. Keyword Co-Occurrence Hierarchical Cluster

Keyword co-occurrence hierarchical cluster is a commonly used method of word clustering which intends to assemble closely related individual keywords to integrate them into a new independent category [51]. The analysis of cluster mirrors the close relation between individual keywords and then further demonstrates the present research hotspots of HCWs+OH. One principle to which keyword clustering analysis shall adhere to is to analyze the frequency of the occurrence of the keywords within one article and to classify closely related keywords.

First, we introduced the 51 × 51 keyword co-occurrence matrix into SPSS and selected the analysis-system clustering-cosine to generate a similarity matrix with the largest cosine value. Then we used R to calculate the dissimilarity matrix based on similarity matrix. Finally, by the Squared Euclidean Distance method, the results of the dissimilar matrix of the keywords were introduced into the SPSS 25.0 for the generating of clustering graph. As shown in Figure 6, the connection distance between cluster was displayed, and four keywords clusters were generated.

Cluster 1 concentrates on HCWs’ physical injuries, working conditions and mental health and includes 26 keywords: health promotion, occupational risks, primary healthcare, health personnel, shiftwork, working conditions, job satisfaction, workplace, training, stress, burnout, epidemiology, healthcare, strength training, backpain, nurses, severe acute respiratory syndrome, mental health, risk assessment, HIV, infection control, blood-borne pathogens, South Africa, TB, safety, and sickness absence.

Cluster 2 incorporates HCWs’ occupational hazards and occupational diseases and includes five keywords: health workers, HCW, occupational hazards, personal protective equipment and occupational diseases. The most extreme categories are Category 1 and Category 2, which indicates that physical injuries, infection diseases, working environment were the most important subjects in the HCWs+OH field.

Cluster 3 encompasses 12 keywords and covers infectious factors, which include: influenza, needle stick injuries, occupational injury, underreporting, occupational health and safety, measles, hepatitis B, surveillance, nursing, OH, vaccination and hospital.

Cluster 4 deals with community health workers and occupational exposure, including eight keywords: musculoskeletal disorders, work, latex allergy, needlestick, community health workers, occupational exposure, surgery and HIV.

### 3.6. Research Trend of HCWs+OH

#### 3.6.1. Document Co-Citation Analysis

Document co-citation analysis (DCA) is a statistical approach to detecting and analyzing research evolution and trends [42]. With the support of CiteSpace, 101,919 references from 402 articles were analyzed. Figure 7 shows the visualization of the document co-citation network containing 191 nodes and 491 links. Each node is labeled with both the name of the first author and the year of publication, which depicts a cited reference [52]. The node size indicates the co-citation frequency of the article. Each link indicates the co-citation relationship between the two articles.

The cluster is marked by indexing terms and a log likelihood ratio (LLR) weighting algorithm. LLR is a computing algorithm which can determine the label of each cluster [53]. As a measurement of the homogeneity or consistency of cluster quality, the maximum nine clusters have a silhouette score above 0.8, which indicates they are close to the highest value of 1.00 and have reliable quality. Size represents the number of articles a given cluster contains. We restricted our analysis to the clusters with size greater than seven to filter small clusters out, and the seven clusters in Table 4 were left.

As shown in Table 4, Cluster 0 indicates the largest cluster with the label “TB”, which includes 32 members, and it reveals “TB” is one of the greatest hot topics in the research of HCWs+OH. There was substantial risk of TB among health care workers [54], and the occupational risk of acquiring TB among HCWs varies remarkably among and within hospitals or institutions [55].

The second is Cluster 1, labeled with “strength training”, which has 27 members. HCWs often perform patient handling, which can cause physical injuries such as musculoskeletal disorders [56]. The third one is Cluster 2 with the tag “influenza” containing 18 members. Influenza causes special hazards in healthcare institutions and can lead to explosive outbreaks of disease, and HCWs were at risk of contracting influenza which also served as an important reservoir for the patients they cared for [57]. Cluster 3 (“HCW”) and Cluster 4 (“occupational exposure”) encompassed 16 members and 13 members respectively. Mucocutaneous exposure to blood, needlestick injuries and secretion of patients’ body fluids are among the most acute threats facing the HCWs [58].

Cluster 5 and Cluster 7, with the tags “epidemiology” and “psychological” encompass 13 members and eight members, respectively. Cluster 5 focuses on infectious diseases such as H1N1 [59], biological-exposure [60], Ebola [61], TB [62], HIV [63]. Among all the health issues, mental health deserves particular attention, since people would not pay too much attention to it. As more workers get recruited in hospitals and healthcare organizations, psychosocial work hazards for HCWs, including physicians, are expected to become worse [64]. In addition, HCWs’ experience of violence and aggression from both physical and non-physical aspects are closely connected to low job satisfaction, high occupational stress, and poor patient care effectiveness [65]. Violence and aggression in the workplace constitutes the third predominant cause of death, which leads to severe problems in safety and health, which can undermine HCWs’ ability to focus on their work [66]. The World Health Organization reported that violence and aggression in the medical profession had a negative impact on employees as well as on workplace, colleagues, families and society, and that exposure to occupational violence and aggression may lead to injury or death [67].

#### 3.6.2. Highly Cited Articles Analysis

Articles with the most citations are usually considered landmark articles in the bibliographic landscape of the subject due to their ground-breaking contributions [52]. From Figure 7 we can clearly see the big nodes in each cluster. Table 5 lists the top 10 reference articles according to their citation frequency from 1992–2019, during which the landmark articles also belonging to their clusters. The top 10 articles are also displayed in Figure 7 with large size nodes and are linked to almost the whole network.

Combining Table 5 and Figure 7, it is notable that one of the landmark articles in Cluster 0 is entitled “High Incidence of Hospital Admissions with Multidrug-Resistant and Extensively Drug-Resistant Tuberculosis Among South African Health Care Workers” authored by O’Donnell MR et al. and published in Annals Of Internal Medicine in 2010; the Impact Factor (five-year) was 19.676, with 120 citations in WOSCC and a 14 times citation frequency among 402 articles. O’Donnell MR et al. [68] concluded that there was a high incidence of hospital admission for drug-resistant TB among HCWs in KwaZulu-Natal, South Africa, due to increased occupational exposure to drug-resistant Mycobacterium TB within health care settings. The other articles from Cluster 0 were published by Baussano I et al. and studied TB among HCWs from different perspectives. Baussano I et al. [69] drew the conclusion that the risk for acquiring TB HCWs suffers is consistently much higher than that of the general mass worldwide. These were followed by articles by Joshi R et al. and Naidoo S et al. which also belong to Cluster 0. Joshi R et al. [3] summarized that it was a necessity to devise and implement a more effective and affordable TB infection-control programs in healthcare facilities especially in countries with low- and middle-income. Naidoo S et al. [70] concluded that, compared with community-acquired TB, the incidence of TB acquired in public sector hospitals among HCWs is much more elevated and it has a tendency to increase every year.

The article published in PLOS ONE entitled “Threshold of Musculoskeletal Pain Intensity for Increased Risk of Long-Term Sickness Absence among Female Healthcare Workers in Eldercare” belongs to Cluster 1; authored by Andersen LL et al. [71], it summarized that pain intensity threshold greatly increased the risk of suffering from long-term sickness among female HCWs. The article belonging to Cluster 2 authored by Menzies D et al. [72] summarized that when the exposure increases and the infection control measures were insufficient, the risk was particularly high.

Two of the top 10 cited frequency articles authored by Yassi A et al. and Alamgir H et al. pertain to Cluster 3. Yassi A et al. [73] conducted a survey in South Africa that highlighted the weaknesses of HCWs in using tool knowledge and suggested the need to improve training. Alamgir H et al. [74] found that direct care occupations have different occupational injury risks depending on their specific tasks and roles in each medical environment.

Remarkably, the article entitled “Estimation of the global burden of disease attributable to contaminated sharps injuries among health-care workers” is the most cited article in Table 5 with 303 citations pertaining to Cluster 4. Pruss-Ustun A et al. [75] concluded that one of the important sources of infections with bloodborne pathogens was occupational exposure and these infections were preventable and should be prevented. An article, published in *Clinical Microbiology Reviews,* had an Impact Factor (five-year) of 24.59, and was entitled “*Risk and management of blood-borne infections in health care workers.*” This had 245 citations, was authored by Alamgir H et al. and was subordinate to Cluster 5. Alamgir H et al. [76] reviewed the risks and management of HIV, HBV and HCV and discussed methods to prevent exposure.

#### 3.6.3. Cluster Timeline Visualization

To clearly identify the developments in HCWs+OH, a timeline visualization was generated based on 10,919 references. The timeline view shows the network by synchronizing its clusters with the horizontal timeline. In the visualization, the line that connects two items indicates co-citation links. The thickness of a line refers to the strength of co-citation. The cited reference is expressed as a circle filled with a citation ring, and its thickness represents the quantity of citation received in the time slice [77]. Therefore, a circle with large size denotes a highly cited reference.

Figure 8 demonstrates a timeline visualization of the seven clusters mentioned and their interior relationships. The earliest is Cluster 4 (occupational exposure), and it reveals that researchers began to pay attention to the HCWs+OH when they first noticed hazard caused by occupational exposure. Then, followed by Cluster 5 (epidemiology) and Cluster 7 (psychological), academics began to carry out research into the influence of epidemiology on HCWs and their mental health from 1979. However, Cluster 5 and Cluster 7 do not have many current high-profile publications. It can be seen from the cluster timeline visualization that Cluster 0 (TB) bears a high assemblage of nodes between 2004 and 2018, which echoes the fact that “TB” is a hotspot in the study of HCWs+OH in recent years. In addition, Cluster 1 (strength training) and Cluster 2(influenza) appear to have recent publications since 2014.

## 4. Discussion

In this study, bibliometric was used for Performance Analysis and Science Mapping, including analysis and evaluation of the publication years, research area distribution, document co-citation network, international productivity and collaboration, highly cited articles and cluster timeline visualization. Time series was used to predict the trend of the publication years. The SNA method was applied to analyze author productivity and collaboration as well as keyword co-occurrence frequency.

Combing bibliometric and time series predictive analysis, we found that both the number and percentage of articles about HCWs+OH among the total articles related to OH in WOSCC indicated an increase trend. From 1992 to 2018, the statistic number of relevant articles raised from 2 to 43 with percentage raising from 1.47% to 3.79%. Moreover, the ARIMA model was used to predict that the publications may reach 87 in 2027, accounting for 4.10% of OH research per year, which means HCWs+OH will receive continuous growing attention in the OH research field.

From research area distribution (Figure 2.), we found that public environmental occupational health prevailed in other areas; this reflected the close relationship between the HCWs+OH and public environmental occupational health. Besides, the distribution of HCWs+OH research areas also suggested the high proneness through infectious diseases, general internal medicine. Furthermore, nursing personnel’s OH enjoyed much more research compared with other HCWs. Medicine and psychology studies also made a great contribution to the HCWs+OH research field. This enables academics working on HCWs+OH research to deduce the research area distribution and further submit papers to journals in the corresponding field.

We used a bibliometric Science Mapping tool to visualize the international collaboration network and found that the top ten countries with the largest number of publications (Figure 3) include America (83), Canada (42), Brazil (39), England (30), France (12), Spain (11), the Netherlands (11), South Africa and China (11), accounting for 67.16% of the total publications. Combined with Table 1, it can be found that Americas accounted for 60.74%, and Europe accounted for 31.11% of the total outputs in the top ten countries and also played vital roles in establishing links with other countries. China published 11 articles but shows inadequate cooperation with other countries. Research on HCWs+OH is a global problem, and countries should strengthen their cooperation with each other. To some extent, we could also draw the conclusion that research on HCWs+OH was more valued in developed countries. In developing countries, only 5% to 10% of workers are accessible to appropriate OH care, while that percentage goes to 20 to 50 among the workers in industrialized countries and it is positively related to the output of scientific research [78].

The SNA method was used to analyze the author co-occurrence network by listing the top 20 most productive authors (Table 2.) and generate a collaboration network between them (Figure 4.). Yassi A (17), who had the most publications, is also placed in the central position in the collaborative network, followed by Alamgir H (9), Spiegel JM(9), Jakobsen MD(8), Andersen LL (8), Bryce EA(8), Brandt M (7), Sundstrup E (7), Yu SC(7), MacIntyre CR (5), Seale H (5), and Alamgir H, Donohue R, Andersen LL, and Seale H; all of them played vital rules in their respective collaborative network. Most of the members of these collaborative networks are from Canada, Australia, South Africa, Denmark, the USA, Brazil, and England. However, there was clear insufficiency of collaboration between authors, which implies a necessity to strengthen the research collaboration and exchange between authors in this field. Besides, academics who do research in HCWs+OH area can also seek cooperation with members of these collaboration networks.

By analyzing keyword co-occurrence frequency and hierarchical clustering, some research hotspots can be discovered. As shown in Table 3 and Figure 5, in addition to “OH”, “HCW”, the hot keywords included “TB”, “infection control”, “HIV”, “vaccination”, “occupational injury”, “occupational exposure”, “mental health” and “working conditions”, “surveillance”, “personal protective equipment”, “sickness absence”, “job satisfaction” etc. Furthermore, four hotspot clusters were generated including: Cluster 1 (physical injuries, workplace, mental health), Cluster 2 (occupational hazards and diseases), Cluster 3 (infectious factors), Cluster 4 (community health workers and occupational exposure). We classified the four hotspot hierarchical clusters into two major categories: injury (Clusters 1 and 2) and exposure (Clusters 3 and 4).

Topic 1 (Clusters 1 and 2) demonstrated the effects of working conditions on HCWs that academics focused on. The working conditions have been seen as a determinant of HCWs+OH due to structural changes. Among the various risks identified in working conditions, such as environmental, psychological, toxicology, and physical risks are challenges to the HCWs+OH research field [79,80]. The psychological issues of HCWs may be caused by work-related risks and poor psychosocial work factors [81]. The European Commission defines the following as psychosocial risks: time pressure, strict hierarchy, lack of satisfaction, support and remuneration; insufficient leadership of personnel, irregular work, social conflicts, violence and aggression, low doctor–patient communication skills. Moreover, occupational violence and aggression have received increasing attention in the field of HCWs & OH research, not only because this is the expectation of social work safety and dignity, but also because the violence has a physiological or emotional impact that exceeds the violent event itself [27].

Topic 2 (Cluster 3 and Cluster 4) reflected exposure risks that academics have paid attention to in recent years. HCWs are classified as the most infectious-disease-prone people [82] Airborne infections render HCWs in a high risky situation [72]. Contaminated or stained sharp objects like needles might bring percutaneous injuries to HCWs considering high exposure under such surroundings [75]. In the 2003 SARS outbreak, HCWs constituted almost half of the confirmed cases in Canada [83]. Similarly, occupational exposure explains nearly 40% of Hepatitis B and C in HCWs [84]. Moreover, exposed in the medical environment, HCWs are confronted with a rather high risk of TB [68] and the much more deadly Ebola as well [85]. This TB risk is elevated among HCWs [86] in South Africa hospitals, where TB outbreaks caused severe death [87] with the HCWs three to ten times more likely to acquire TB as per some studies [88].

Bibliometric Science Mapping was used to analyze DCA, highly cited articles, cluster timeline visualization for detecting the evolution and trends of HCWs+OH research. A total of 101,919 references from 402 articles were analyzed in this section. According to document co-citation analysis, seven clusters were generated (Figure 7), including Cluster 0 (TB), Cluster 1 (strength training), Cluster 2 (influenza), Cluster 3 (HCW), Cluster 4 (occupational exposure), Cluster 5 (epidemiology), and Cluster 7 (psychological); besides, the top 10 most frequently cited articles were detected as well (Table 5). From the analysis of top ten most frequently cited articles, we can find the influential articles, authors and literature resources in HCWs+OH field. Of the top 10 frequently cited articles, five were concerned with the transmission of TB among HCWs, and other articles were about working conditions, physical injuries and other infectious diseases for HCWs. Influential authors included O’Donnell MR, Pruss-Ustun A, Baussano I, Joshi R, Naidoo S, etc. The sources of articles with more citations and higher influential factors included *Clinical Microbiology*, *Annals Of Internal Medicine, PLOS Medicine, Emerging Infectious Diseases,* etc. These provide the scholars studying HCWs+OH with the ideas of these authors and would also offer inspiration for cooperation and article publishing.

Combining with document co-citation analysis with cluster timeline visualization (Figure 8), we detected that the earliest research cluster of HCWs+OH was “occupational exposure”, then followed by “epidemiology” and “psychological”. Notably, the high concentration of nodes between 2004 and 2018 echoes the fact that “TB” has become a hotspot in HCWs+OH field and will continue to be studied in the future. Moreover, in recent years, research on “strength training” and “influenza” has gradually increased and it may attract a lot of attention in the future.

Although researchers and practitioners have made substantial achievements in the promotion of HCWs+OH theory and practice, some shortcomings deserve due attention. (1) Lack of comprehensive and systematic research on the evaluation system of HCWs+OH. Most of the existing HCWs+OH evaluation systems are based on a single factor, disease, or aim to carry out statistical or investigative research on the current situation of OH, which inevitably lacks in general applicability, so a comprehensive and systematic study of the HCWs+OH evaluation system is needed. (2) Lack of research on the information system of HCWs+OH. Although information systems are widely used in health sectors around the world, few initiatives have been made for HCWs+OH [89]. The implementation and application of information systems is a resource-intensive undertaking and needs to be elaborated on in the future. (3) Research on HHCWs should receive more attention. In view of the aging of the population, the increasing cost of chronic diseases and health care, more people need to be treated at home, which increases the demand for more HHCWs [30]. HHCWs may also be injured by musculoskeletal disorders [90], stress [33], exposure [91], etc. in the process of work, but existing research is far from a clear understanding of the OH of HHCWs; therefore, clear research on the OH of HHCWs is necessary.

## 5. Conclusions

This study exhibits a detailed overview of HCWs+OH research with bibliometric and SNA methods, covering related articles from WOSCC form 1992–2019. Bibliometric and SNA methods were adopted to map and quantify publication trends, research area distribution, international productivity and collaboration, keyword co-occurrence frequency and hierarchical clustering, highly cited articles and cluster timeline visualization. According to bibliometric and time series predictive analysis, we found that the research on HCWs+OH gained increasing attention since 1992. The research area distribution showed the high priority of public environmental and occupational health, general internal medicine, infectious diseases, nursing, microbiology and psychology. The international collaboration network revealed the top ten countries with the largest number of publications (Figure 3.) including America (83), Canada (42), Brazil (39), England (30), France (12), Spain (11), the Netherlands (11), South Africa and China (11) accounted for 67.16% of the total publications. The most-contributing author was Yassi A; however, there was deficiency in cooperation between each author.

By analyzing keyword co-occurrence frequency and hierarchical clustering, four hotspot clusters were generated including: Cluster 1 (physical injuries, workplace, mental health), Cluster 2 (occupational hazards and diseases), Cluster 3 (infectious factors), Cluster 4 (community health workers and occupational exposure). An analysis of DCA, highly cited articles, cluster timeline visualization, the paper detected the evolution and trends of HCWs+OH. Of the top 10 frequently cited articles, five were concerned with the transmission of TB among HCWs, and other articles were researching working conditions, physical injuries and other infectious diseases for HCWs. According to the document co-citation network (Figure 7.), seven clusters were particularly analyzed. Combined with cluster timeline visualization (Figure 8.), this article held that the earliest research cluster is “occupational exposure” during the periodical studies, followed by Cluster 5 (epidemiology) and Cluster 7 (psychological). However, in recent years, Cluster 0 (TB), Cluster 1 (strength training) and Cluster 2 (influenza) appeared to enjoy more publications.

Through comprehensive bibliometric and SNA methods, this study systematically analyzed the current status of HCWs & OH research, which lays several foundations for future research. Specifically, quantitative analysis of article publication trends, research area distribution, international collaboration networks, author collaboration networks, and highly-cited documents can provide guidance for scholars to select target journals and partners. In addition, it allows relevant scholars to understand the research hotspots and trends of HCWs+OH, which would provide the scholars with an overview of this field as well as contributing to inter-disciplinary studies.

Although combined efforts to systematic and comprehensive approach to collect and analyze relevant articles were made in this study, there were still several limitations. Firstly, the analytical bibliographies of articles were only retrieved from the one mainstream database WOS and excluded some biomedical database such as PubMed and Scopus. Additionally, it is likely that some related article might have been overlooked, due to the limitations of software applicability. In addition, although the retrieved articles can represent most opinions in HCWs+OH research, some valuable articles published in other formats might be inevitably neglected. Finally, when performing bibliometrics and SNA methods for statistics, the results of software statistics have a certain deviation. Further study will use more accurate software and search more databases as well as adopting different terms such as MeSH to improve the research results, and a systematic investigation containing valuable reports, letters and books is expected to draw a more comprehensive knowledge map for future research.

## Figures and Tables

**Figure 1 ijerph-17-02625-f001:**
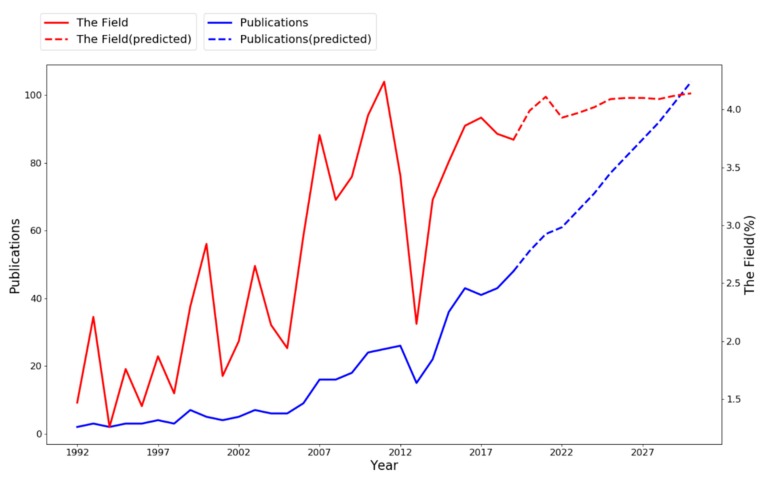
The publications on HCWs+OH and the number of articles on HCWs+OH/total articles in the field of OH.

**Figure 2 ijerph-17-02625-f002:**
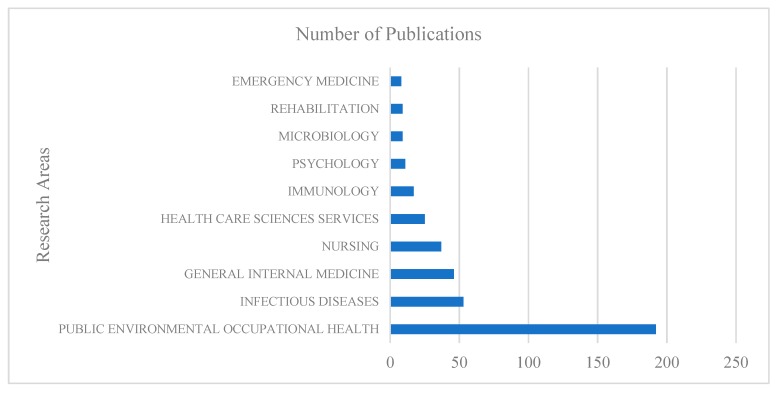
Research area distribution of HCWs+OH in WOSCC.

**Figure 3 ijerph-17-02625-f003:**
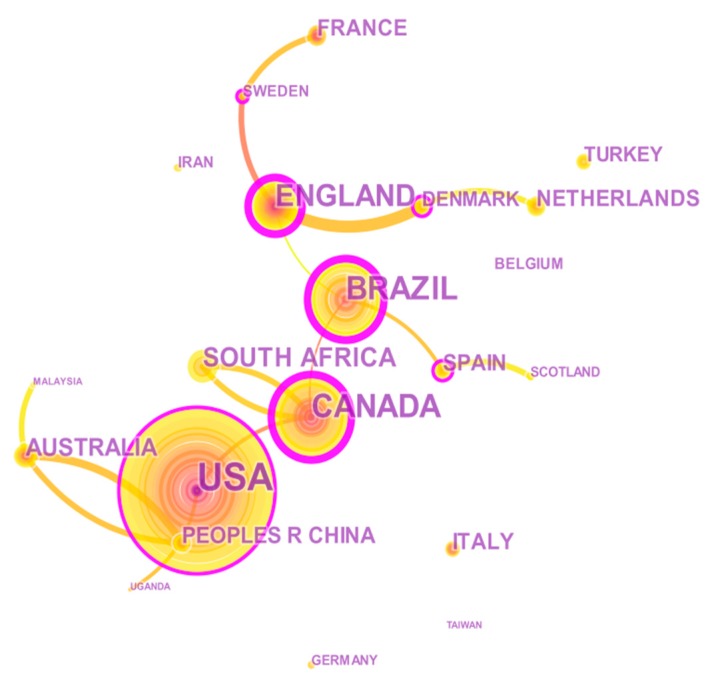
International collaboration network.

**Figure 4 ijerph-17-02625-f004:**
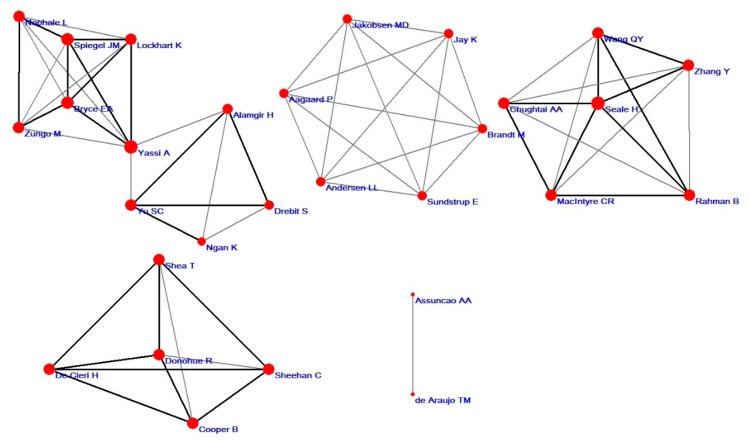
The top 20 most productive authors collaboration network.

**Figure 5 ijerph-17-02625-f005:**
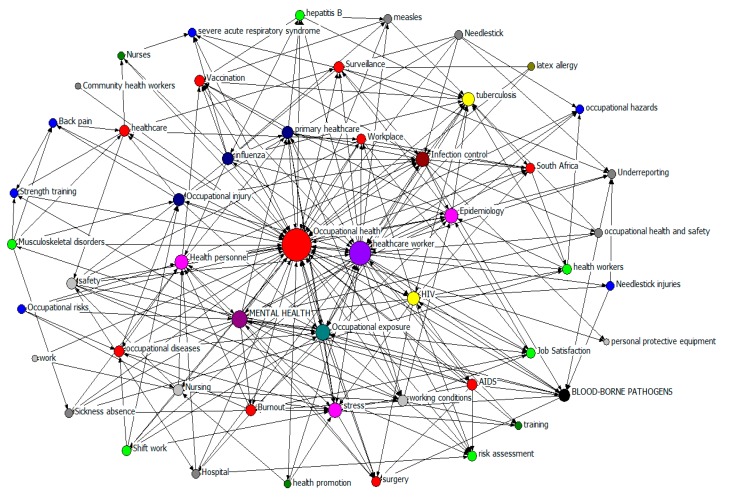
Keyword co-occurrence network of the 51 most frequent keywords by degree.

**Figure 6 ijerph-17-02625-f006:**
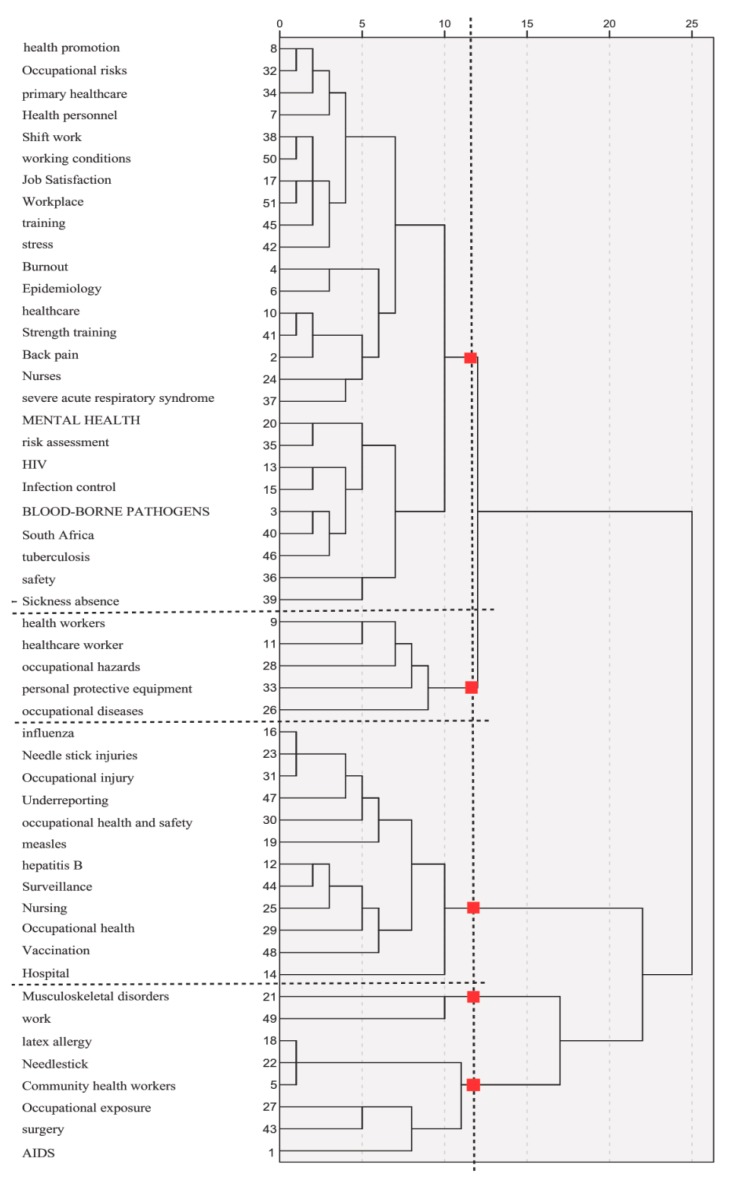
Keyword co-occurrence hierarchical cluster graph for the field of HCWs+OH research.

**Figure 7 ijerph-17-02625-f007:**
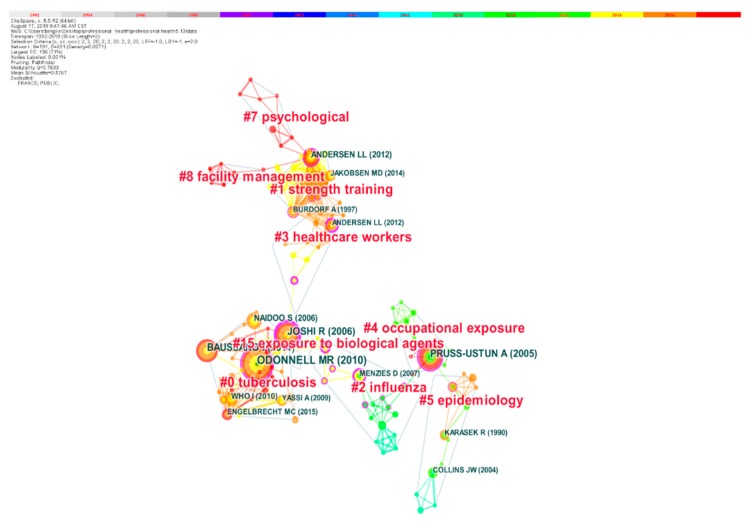
Document co-citation network.

**Figure 8 ijerph-17-02625-f008:**
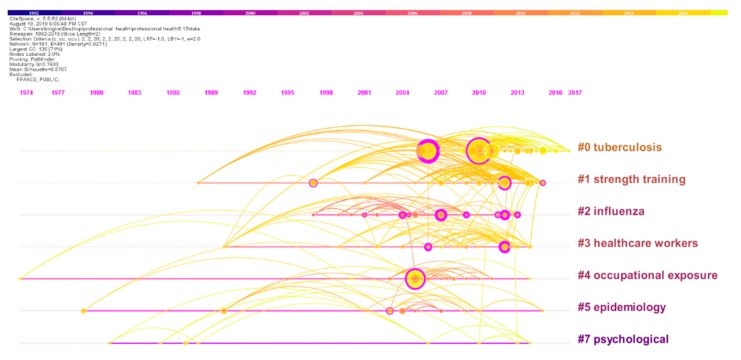
Document co-citation analysis clusters timeline visualization.

**Table 1 ijerph-17-02625-t001:** Top ten countries with the largest number of publications.

Country	Territory	Frequency	BC	Country	Territory	Frequency	BC
USA	North America	83	0.41	AUSTRALIA	Oceania	13	0.17
CANADA	North America	42	0.33	FRANCE	Europe	12	0.02
BRAZIL	South America	39	0.31	SPAIN	Europe	11	0.15
ENGLAND	Europe	30	0.13	NETHERLANDS	Europe	11	0.13
SOUTH AFRICA	AFRICA	18	0.06	PEOPLES R CHINA	Asia	11	0

**Table 2 ijerph-17-02625-t002:** Different centrality measures for top 20 most productive authors.

Ranking	Authors	# of Papers	DC	BC	CC
1	Yassi A	17	7	0.05	0.28
2	Alamgir H	9	4	0.02	0.22
3	Spiegel JM	9	5	0.00	0.21
4	Jakobsen MD	8	5	0.00	0.21
5	Andersen LL	8	5	0.00	0.21
6	Bryce EA	8	5	0.00	0.21
7	Brandt M	7	5	0.00	0.21
8	Sundstrup E	7	5	0.00	0.21
9	Yu SC	7	4	0.02	0.22
10	MacIntyre CR	5	5	0.00	0.21
11	Seale H	5	5	0.00	0.21
12	Jay K	4	5	0.00	0.21
13	Aagaard P	4	5	0.00	0.21
14	Zungu M	4	5	0.00	0.21
15	Rahman B	4	5	0.00	0.21
16	Lockhart K	4	5	0.00	0.21
17	Zhang Y	3	5	0.00	0.21
18	Sheehan C	3	4	0.00	0.17
19	Donohue R	3	4	0.00	0.17
20	Wang QY	3	5	0.00	0.21

**Table 3 ijerph-17-02625-t003:** Top 22 keywords ranked by counts.

Ranking	Keyword	Counts	Ranking	Keyword	Counts
1	OH	152	12	influenza	11
2	HCW	85	13	Nursing	10
3	TB	20	14	healthcare	10
4	infection control	18	15	health workers	9
5	Health Personnel	17	16	Mental Health	8
6	HIV	16	17	epidemiology	8
7	occupational injury	14	18	Back pain	7
8	Occupational Exposure	13	19	health promotion	7
9	vaccination	13	20	Musculoskeletal disorders	7
10	Primary Healthcare	12	21	burnout	7
11	Stress	12	22	working conditions	7

**Table 4 ijerph-17-02625-t004:** Seven largest clusters sorted by size.

Cluster #	Label	Size	Silhouette	Mean (Year)	Top 5 Terms (Log-Likelihood Ratio)
0	TB	32	0.951	2012	TB (16.23); health worker (8.42); HIV (4.86); South Africa (4.19); cluster randomized controlled trial (4.19)
1	strength training	27	0.92	2009	strength training (20.64); back pain (12.4); musculoskeletal disorders (7.7); patient handling (6.77); shoulder pain (6.77)
2	influenza	18	0.947	2005	influenza (9.54); swine origin influenza (4.74); H1N1 (4.74); delivery of health care (4.74); paramedics (4.74)
3	healthcare worker	16	0.877	2006	HCWs (5.38); low back injury (3.54); injury (3.54); hierarchical rating method (3.54); shift work (3.54)
4	occupational exposure	13	0.99	2004	occupational exposure (9.76); health personnel (9.76); HCW safety (5.33); needlestick injuries (5.33); blood-borne infections (5.33)
5	epidemiology	13	0.992	2003	epidemiology (7.55); mental disorders (7.55); working conditions (4.06); evaluation (3.76); stress psychological (3.76)
7	psychological	8	0.984	1998	psychological (4.68); work capacity evaluation (4.68); health care workers (4.68); reproducibility of results (4.68); sickness presenteeism (4.68)

**Table 5 ijerph-17-02625-t005:** Top 10 Most Frequently Cited Articles During 1992–2019.

CF	Author (Year)	Literature Sources	Volume (Issue): Pages	TC	IF	#
14	O’donnell MR (2010) [68]	ANNALS OF INTERNAL MEDICINE	153(8):516–522	120	19.676	0
11	Pruss-ustun A (2005) [75]	AMERICAN JOURNAL OF INDUSTRIAL MEDICINE	48(6):482–490	303	2.089	4
10	Baussano I [69]	EMERGING INFECTIOUS DISEASES	17(3):488–494	157	7.152	0
10	Joshi R (2001) [3]	PLOS MEDICINE	3(12): 2376–2391	279	14.814	0
7	Naidoo S (2006) [70]	INTERNATIONAL JOURNAL OF TB AND LUNG DISEASE	10(6):676–682	51	2.363	0
7	Yassi A (2009) [73]	INTERNATIONAL JOURNAL OF OCCUPATIONAL AND ENVIRONMENTAL HEALTH	15(4):360–369	14	1.756	3
7	Menzies D (2007) [72]	INTERNATIONAL JOURNAL OF TB AND LUNG DISEASE	11(6):593–605	262	2.363	2
7	Andersen LL (2012) [71]	PLOS ONE	7(7): e41287	52	3.337	1
6	Beltrami EM (2000) [76]	CLINICAL MICROBIOLOGY REVIEWS	13(3):385–407	245	25.599	5
5	Alamgir H (2007) [74]	OCCUPATIONAL AND ENVIRONMENTAL MEDICINE	64(11):769–775	47	4.062	3

Notes: CF= Cited Frequency among 402 articles; TC=Times Cited in Web of Science Core Collection; IF= Impact Factor (Five year); # = Cluster number.

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
