# Peer review of "Mapping the Scientific Research on Healthcare Workers’ Occupational Health: A Bibliometric and Social Network Analysis"

_ijerph, 2020, doi:10.3390/ijerph17082625_

Round 1

Reviewer 1 Report

This is an interesting manuscript and provides a bibliometric analysis about occupational health of healthcare workers. In general, this paper is well written, I have only some observations:

  • Line 138: “. 2007 demarcated…”. It is better no to initiate a paragraph with numbers, but letters. So, I suggest to paraphrase or rewrite this sentence, for example: “The 2007 year, demarcated as the debut …” or “Two thousand seven year,…”
  • All references must be checked and corrected according to the journal style:
    • References 46 and 49 are the same “Kumar et al., but written in different ways.
    • Cite journals according to ISO Abbreviation system. For example: Reference 2, instead of “Annals of Agricultural and Environmental Medicine”, it would be better to cite as: “Ann Agric Environ Med”.
  • One issue that the authors did not directly take into account —it was considered in Introduction and References, though—, but, is currently a matter of concern is violence and aggression towards health care professionals. I strongly believe the authors should address this very important topic in their analysis.

Author Response

Thank you very much for your comments. We responded to your suggestions point-by-point, please see the attachment.

Reviewer 2 Report

1. It is understood that some useful outcomes were obtained from the research. However the authors needs to spell out which parts of the result can be linked up clearly with the models.

2. While a high level of research work was seen, there are quite a large number of typos and grammatical errors noted.

3. It is desirable to have a list for the abbreviations and acronyms used in the journals before publishing.

Author Response

(The authors gave the same response as above.)

Reviewer 3 Report

This article is interesting to read. Most analyses were well done. However, I have the following concern on publication trend: I would recommend the authors use a relative measure (i.e. number of articles on healthcare workers' occupational health/total articles in the field of occupational health) to indicate how popular this topic was in the past three decades.In addition, the authors had better predict the trend using machine learning/time series methods (for example, like what these authors have done: https://doi.org/10.1371/journal.pone.0199510). 

In addition, the authors had better provide in-depth discussion on the research landscape of HWOH and tell readers what can be taken away from this article. Especially, which topics would be more popular in the future and what are the gaps in the existing literature. 

Hope my comments are helpful.  

Author Response

(The authors gave the same response as above.)

Round 2

Reviewer 1 Report

I have read carefully this new version of the manuscript, and I am completely satisfied with the answers to my previous observations. I have no more comments for the authors.

Reviewer 3 Report

The authors have made significant revision and I am satisfied with their work.